# GAN-WGCNA: Calculating gene modules to identify key intermediate regulators in cocaine addiction

Taehyeong Kim[1]☯, Kyoungmin Lee[1]☯, Mookyung Cheon[2]*, Wookyung Yu[1]*

1 Department of Brain Sciences, Daegu Gyeongbuk Institute of Science & Technology, Daegu, South Korea,
2 Dementia Research Group, Korean Brain Research Institute, Daegu, South Korea

☯ These authors contributed equally to this work.
* mkcheon@kbri.re.kr (MC); wkyu@dgist.ac.kr (WY)

**Data Availability Statement:** All the data and codes underlying this article are available in Github (https://github.com/baicalin/GAN-WGCNA) and Gene Expression Omnibus (GSE110344).

## Abstract

Understanding time-series interplay of genes is essential for diagnosis and treatment of disease. Spatio-temporally enriched NGS data contain important underlying regulatory mechanisms of biological processes. Generative adversarial networks (GANs) have been used to augment biological data to describe hidden intermediate time-series gene expression profiles during specific biological processes. Developing a pipeline that uses augmented time-series gene expression profiles is needed to provide an unbiased systemic-level map of biological processes and test for the statistical significance of the generated dataset, leading to the discovery of hidden intermediate regulators. Two analytical methods, GAN-WGCNA (weighted gene co-expression network analysis) and rDEG (rescued differentially expressed gene), interpreted spatiotemporal information and screened intermediate genes during cocaine addiction. GAN-WGCNA enables correlation calculations between phenotype and gene expression profiles and visualizes time-series gene module interplay. We analyzed a transcriptome dataset of two weeks of cocaine self-administration in C57BL/6J mice. Utilizing GAN-WGCNA, two genes (Alcam and Celf4) were selected as missed intermediate significant genes that showed high correlation with addiction behavior. Their correlation with addictive behavior was observed to be notably significant in aspect of statistics, and their expression and co-regulation were comprehensively mapped in terms of time, brain region, and biological process.

## Introduction

Complex time-series gene interactions can be observed in various biological processes, such as the cell cycle [1–3], circadian rhythm [4, 5], development [6], and pathogenesis. One such example is the regulation of synaptic plasticity in the nervous system. The functionality of key factors at specific spatiotemporal points—where, when, and how—is important to this process. Indeed, the disturbance of a key factor in different spatiotemporal manners can induce different diseases. As shown in these biological processes, understanding the spatiotemporal

**Funding:** This work was supported by the RandD programs of DGIST (22-CoE-BT-01), funded by the Ministry of Science and ICT of Korea. This research was supported by the KBRI Basic Research Program through the Korea Brain Research Institute, funded by the Ministry of Science and ICT (22-BR-02-04 [MC]) and the National Research Foundation of Korea (NRF) grant funded by the Korea government (MSIT) (2021R1A2C1003657 [TK,KL,MC] NRF-2023R1A2C1006248(WY)). The funders had no role in study design, data collection and analysis, decision to publish, or preparation of the manuscript.

**Competing interests:** The authors have declared that no competing interests exist.

**Abbreviations:** GAN, Generative adversarial network; WGCNA, weighted gene co-expression network analysis; rDEG, rescued differentially expressed genes; Alcam, Activated Leukocyte Cell Adhesion Molecule; Celf4, CUGBP Elav-like family member 4; NGS, Next Generation Sequencing; PFC, Prefrontal cortex; DStr, Dorsal striatum'; CPU, Caudate putamen; NAc, Nucleus accumbens; BLA, Basolateral amygdala; vHIP, Ventral hippocampus; VTA, Ventral tegmental area.

interplay of key factors is essential for the diagnosis and treatment of diseases, particularly those with spatiotemporal complexity.

To understand the spatiotemporal complexity, several experimental methods have been devised. One experimental approach is systems biology, particularly -omics data generation using sequencing technology. Typical single-cell RNA sequencing (scRNA-seq) data, which reads gene expression levels of each cell, consist of 20,000–30,000 cells with over 10,000 genes. Meanwhile, bulk RNA sequencing reads gene expression levels of each tissue sample, usually up to a few hundred. Basic interpretation can be done using Differentially Expressed Gene (DEG) calculations from bulk RNA sequencing, which typically identify over one hundred significant genes. In most biological studies using these hundred components, research should involve complex gene regulatory machinery. Although the network is spatiotemporally complex, the current pipeline in most cases provides only weak or no spatiotemporal regulatory information.

One of the hurdles for the thorough utilization of transcriptomic next-generation sequencing (NGS) data is the insufficient degree of sample data density, likely due to the high cost associated with various spatial, temporal, and treatment conditions. The limited availability of biological samples is often described as large features (genes) and small samples that cause overfitting and high-variance problems.

To overcome or compensate for expensive experimental methods, several in silico approaches have been devised. For example, Marouf et al. demonstrated that generating real-like omics data with non-linear properties is possible using advanced computational techniques [7]. However, simply generating more data may not be sufficient for credibility and adding insights to previous studies. Instead, unveiling a hidden trajectory of the transcriptome between the initial and final phases of experimental conditions affected by disease or drugs with finer time resolution could provide a better lens to observe minute details of pathogenesis-related gene modules without any additional experiments

Generative Adversarial Network (GAN) may be a solution for this problem. A GAN has been used in several domains, including image processing, not only to achieve a higher performance generative model with a lesser amount of training data [8–12], but also to provide unique operations such as the generation of semantically meaningful fake data using latent space interpolation. Interpolation between generative data by latent vector arithmetic generates intermediate fakes that seem authentic and realistic, suggesting that the latent space of GAN encodes feature space information. Ghahramani *et al.* (2018), used this idea in biological research and achieved an increase in the temporal resolution of gene expression data using single-cell RNA-seq (scRNA-seq) data from epidermal cells [13]. Inspired by this idea, Park *et al.* (2020) showed that increased temporal sample density can be used in biological research related to Alzheimer's disease using only augmented bulk mRNA-seq data [14]. The training procedure and validation used in this research followed Jinhee Park's work with minor adjustments.

Accompanying with GAN, Weighted Gene Co-expression Network Analysis (WGCNA) was used in this study which is a powerful systemic biology method used to describe the correlation patterns among genes across multiple samples. This method calculates a network based on gene expression data, where nodes represent genes and edges reflect the strength of the correlation between gene pairs. By grouping genes with similar expression patterns into modules, WGCNA allows researchers to identify clusters of co-expressed genes that may share common biological functions or regulatory mechanisms. WGCNA is one of well-accepted conventional methods that particularly valuable for exploring complex biological processes and diseases, as it integrates high-dimensional gene expression data to study gene networks associated with specific phenotypes or conditions [15–19].

In this study, we devised two analytic methods: GAN-WGCNA (weighted gene co-expression network analysis) and rDEG (rescued differentially expressed genes). GAN-WGCNA is a combination of the GAN and WGCNA methods that calculates co-expressed gene modules from generated time-series data at the systemic level. The calculated modules can be used to analyze their correlation with features or to visualize module networks, which are accompanied by conventional WGCNA. rDEG is also an adapted method for gene-level analysis based on the conventional DEG method. By calculating the DEG in the generated time-series data, the rDEG rescue the missed DEG in intermediate stages.

Using these two methods, we tackled the ambiguity of the spatiotemporal interplay of cocaine addiction, which is a good example of a biological process that has spatiotemporal complexity [20, 21]. It shows not only complex gene expression levels but also sophisticated interplay between brain regions. This spatio-temporal interplay leads to different symptoms, such as coronary artery disease [22], vascular disease [23], and even mortality [24]. Several studies have used conventional methods for transcriptome analysis of cocaine addiction [25–27]. Using the same data as previous research but with different methods, we showed that gene modules have high correlations with cocaine addiction behavior. They consisted of two distinctive gene module networks, which were also distinguished by Gene Ontology analysis.

## Methods

### C57BL/6J cocaine self-administration dataset

We used a public dataset generated by a previous study [28] and followed their RNA-seq analysis method. The detailed method is described in S1 File.

### Voom and data preprocessing for training

After sequencing data processing, we also followed normalization and augmentation methods which applied in previous studies [14, 28]. Detailed method is described in S2 File.

### Wasserstein generative adversarial networks with gradient penalty loss

We used the Wasserstein generative adversarial network and gradient penalty loss (WGAN-GP) [29] for model training. The Gulrajani, et al suggested following loss for training.

$$\min_{G} \ \max_{DED} E_{x \sim \mathbb{P}_r}[D(x)] - E_{\tilde{x} \sim \mathbb{P}_g}[D(\tilde{x})]$$

$$L = E_{\tilde{x} \sim \mathbb{P}_g}[D(\tilde{x})] - E_{x \sim \mathbb{P}_r}[D(x)] + \lambda E_{\hat{x} \sim \mathbb{P}_{\hat{x}}}\left[\left(||\nabla \hat{x}D(\hat{x})||_2 - 1\right)^2\right]$$

Although we followed most of the previous studies' implementations, including algorithms, pipelines, and hyperparameters, some implementations have been modified as needed. For example, we used the RMSprop optimizer instead of the Adam Optimizer. These modifications were confirmed to meet the fake data generation criteria, with a Pearson correlation value of 0.95. We have documented the other hyperparameters in S1 Fig.

### Averaged gene expression simulation with latent space interpolation

Before generating intermediate transcriptomes between the two datasets, the fakes of each sample data point should be obtained. A resembled fake is not only a real-like fake data paired with its latent-space vector (z) by meeting its criteria, but also has a stable manifold in latent space by averaging multiple latent vectors. To generate a resembled fake corresponding to the sample data, which is an average over multiple z of each sample, we generated 35,000 fakes

using a trained generator (G) and selected 10 nearest latent vectors based on the Pearson correlation value between G(z) and a target real sample. The entire process was handled using the NumPy library in Python [30] and the minimum threshold of the Pearson correlation was 0.95. If 35,000 fakes were unable to generate resembled fakes for each real sample, this process was repeated until a successful resembled fake generation was obtained using different NumPy seed values.

$$\mathbf{delta} = \mathbf{z}_{CN} - \mathbf{z}_{SN}$$

$$\boldsymbol{z}_{i^{th}} = \mathbf{z}_{SN} + \mathbf{delta} * \frac{i}{100} \, (i \text{ is } 1 \text{ to } 99)$$

The delta vector (delta) was created by subtracting the latent vectors between the resembled fakes of the cocaine-addicted state (CN) and control (SN). Subtraction was performed for every possible combination of the resembled fakes of the CN and SN. Next, we generated transcriptomes by using $G(z_i)$. $z_i$ is an intermediate latent vector and $i$ is stated for sequential changes in time. The delta and intermediate latent vectors $z_i$ were constructed using the above equation. This process creates transcriptomes that have several gene expression transition curves during physiological changes from the normal to the addictive state. Finally, by repeating this process ten times at different epochs (10–20 k) and averaging the gene expression transition curves, we created the average gene expression profile at intermediate states.

Generating the average gene expression profile using GAN differs from simple linear interpolation of expression data. This is because the interpolation process occurs only in the latent space, which is mapped into a higher feature space through the trained generator. Consequently, the transition in the feature space is not only distinct from linear interpolation in the feature space but also appears more natural. These differences stem from the fact that the generator learns about the data distributions, capturing higher-level features and structures. Indeed, our generation results reveal various transition curves.

## Weighted gene co-expression network analysis

To utilize complemented information in aspects of systems biology, WGCNA was performed on the averaged gene expression profile using the R library (WGCNA) [31]. Detailed process which followed author provided tutorial is described in S3 File.

## Gene ontology

For high-level interpretation of the constructed module, Gene ontology (GO) analysis was performed on each gene module using the R library *GOstats* [32]. Based on GO enrichment results, each module has a primary GO term which is the most significant GO term using *hyperGTest* and conventional threshold parameters– 0.05 for pvalueCutoff and 50 for cutoff_size. The primary GO term for each module was selected as the first GO term of this cutoff result. However, some GO terms were manually selected. The full results are described in the supplementary material.

## Rescued DEG

An extended version of rDEG was constructed for statistically significant gene expression profile interpretation. By handling the pre-averaged gene expression profile as a biological replica in conventional DEG, we can calculate Log2FoldChange and P-value using r library *voom-limma* [33, 34]. This approach provides quantitative and statistical criteria for genes that are

significant in an intermediate state. A Log2FC value of 0.2 and a P-value of 0.05, were selected as thresholds, which were the same as those in the previous study [28].

### Ethics approval and consent to participate

Not applicable

## Results

### Generation of simulated gene expression profile during cocaine addiction using GAN

In a previous study [28], transcriptome-wide regulation was evaluated in six reward-associated brain regions (PFC, DStr: CPU, NAc, BLA, vHIP, VTA) by comparing DEG analysis in different contexts (acute, re-exposure, etc.) and combining the behavioral addiction index. Based on pattern analysis, they assessed differentially regulated gene expression across brain regions and predicted 192 upstream regulators of cocaine addiction. The method, which is a pairwise DEG comparison of the control and treatment used in the previous study, can be extended using GAN in terms of temporal resolution.

To acquire time-series gene expression profiles that enable more in-depth analysis than previous studies, we applied the GAN method to the cocaine-SA data to create simulated gene expression profiles of the mouse brain transcriptome treated with two-week cocaine-SA. The GAN method uses bulk mRNA-seq data as the training input and generates a time-series intermediate transcriptome using latent space interpolation. Detailed methods and additional information on the data structure during the creation of a simulated dataset and the overall research workflow are presented in S1A–S1C and S2A–S2F Figs.

### Spatiotemporally augmented gene expression profile used in WGCNA

The generated gene expression profile provided enriched spatiotemporal information. We attempted to visualize the gene expression profiles of some important genes involved in cocaine addiction. We visualized one of the well-known cocaine addiction-related genes, Creb. (S3 Fig)

The expression profile of the Creb family shows diverse properties of spatiotemporal gene expression changes, revealing subfamily-specific profile patterns. The expression profile of the Creb 1 gene shows limited expression level changes during cocaine addiction, compared to the Creb3 subfamily, which shows not only significant expression changes but also regional differences. In terms of regional differences, Crebl2 had a homogeneous decreasing expression profile across all regions, and Creb5 showed only valid expression changes in the VTA region. This spatiotemporal information can provide a possible explanation at the transcriptome level for specific biological events, such as sophisticated neuronal plasticity regulation during cocaine addiction [35].

After checking the spatiotemporal information of the generated gene expression profiles, WGCNA was used for quantitative interpretation of extended gene expression information. Conventional WGCNA usually takes the normalized gene expression level as input (i.e., the number of samples and the number of genes), whereas GAN-WGCNA takes the shape of the simulated timesteps and the number of genes. Genes were then grouped into modules based on their profile similarity. Similar genes were grouped into the same module and were considered co-expressed genes.

## GAN enables a calculation of correlation between gene module and behavioral data

By re-calculating the module's eigengene expression profile in real samples, we can combine a gene module that is calculated from a time-series generated sample and behavioral information that is available in real samples (addiction index). This combined information provides us with an opportunity to interpret a feature without temporal information as a feature with a temporal aspect. The addiction index is an indication of addictive behavior and is available in a real sample to detect addiction-associated modules. Calculating the Pearson correlation of the module's eigengene and behavioral data leads to a quantitative correlation between the gene module and cocaine addiction in spatiotemporal aspects. (Figs 1A, S4A, S4B and S5)

The most highly correlated gene modules with addictive behavior were found in the NAc and VTA regions (Fig 1B and 1C), which is consistent with the results of previous studies [26, 36–39]. To calculate the primary biological processes of the highly correlated gene modules, we performed GO analysis. For each of the 51 modules, one representative GO term was selected and presented in S1 Table.

Representative GO terms provide an overview of the biological processes involved in cocaine addiction. This overview can be summarized based on both the module correlation statistics (correlation coefficients, P-values) of addictive behavior and the representative GO term statistics of each module (enrichment ratio, OddRatio, and P value of gene ontology analysis) (Fig 2). According to our criteria, most of the significant gene modules appeared to be located in the NAc region. For the negatively correlated modules, the GO annotations were adhesion of symbiont to host (Module 38), miRNA loading onto RISC involved in gene silencing by miRNA (Module 39), and deoxyribonucleoside metabolic process (Module 40) during cocaine addiction. For the positively correlated modules, information related to the regulation of ventricular cardiac muscle cell membrane repolarization (Module 20) and positive regulation of epithelial cell differentiation (Module 24) is provided.

## Temporal alignment of highly addiction related modules shows a distinctive correlation pattern between cell migration and membrane-related biological process during cocaine addiction

To provide a gene module network in a spatiotemporal manner and seek missed intermediate regulator genes, highly addiction-correlated modules were visualized using temporal networks based on their eigengene expression profiles. (Figs S6 and 3A–3C) Each module was aligned to a specific time point where its eigengene expression level is maximized. Module-to-module connections were visualized using Pearson correlation values between modules, which referred to the module distance in the WGCNA.

In Fig 3, modules with the highest eigengene expression levels on day 0 are placed at the top, and modules that have the highest eigengene expression levels in the intermediate modules are placed in the middle and late modules are placed at the bottom. The network in the early mid–late group shows a distinctive pattern. According to the GO results, one of the intermediate modules showed possible intermediate regulatory functions. Heparan sulfate-related regulation (Module 49) is known as a therapeutic target for cocaine addiction [40, 41]. In previous studies, there was no detailed information regarding how and when they worked. However, through this inspection, we can now estimate when they are activated and how heparan sulfate proteoglycan (HSPG) in the NAc is regulated.

Distinctive correlation patterns were also observed from day 0 to day 14. Based on the correlation pattern, positive regulation of the inositol phosphate biosynthetic process (Module 45) and polysaccharide biosynthetic process (Module 46) in the NAc leads to HSPG biosynthesis

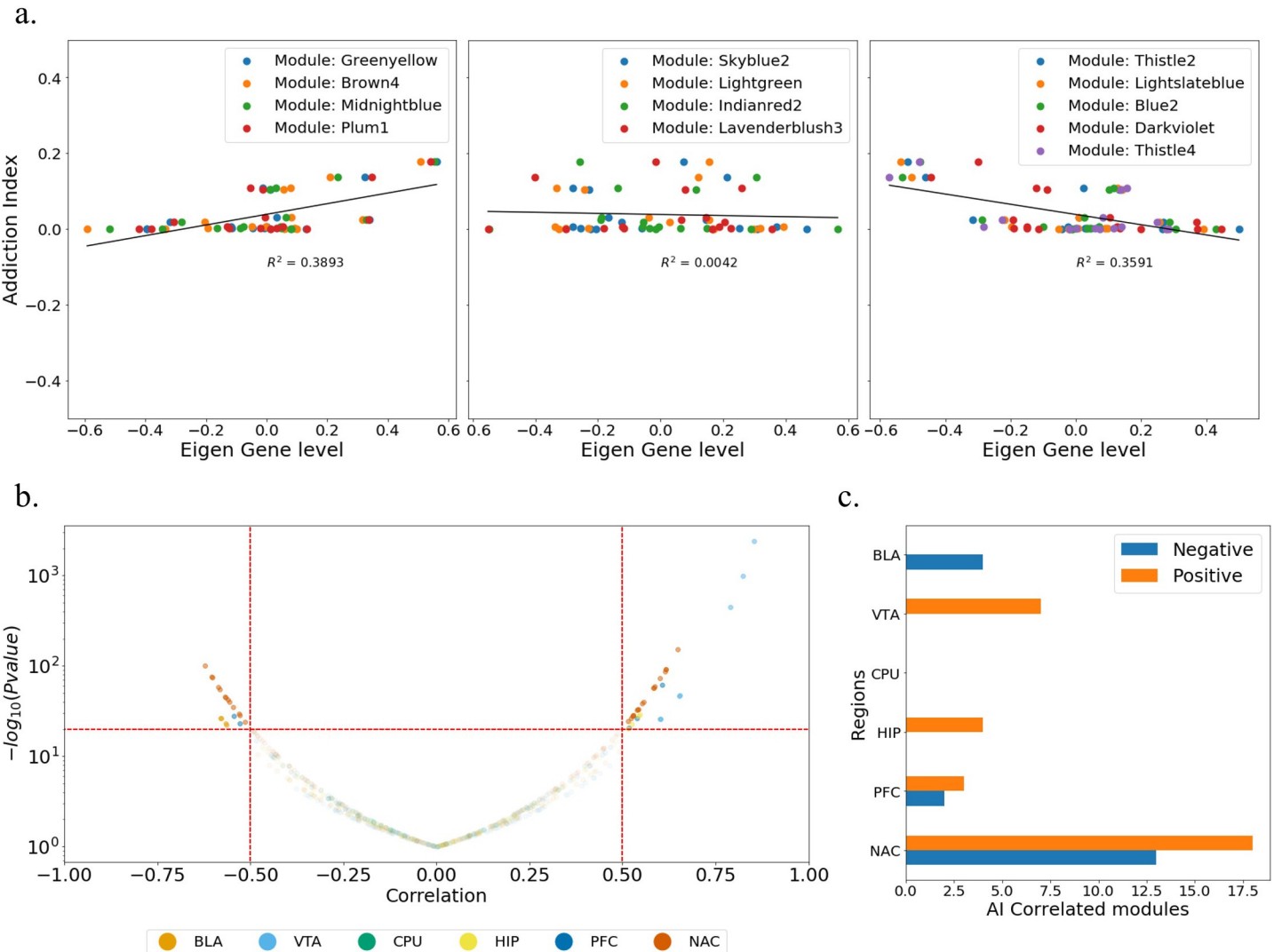

**Fig 1.** Module-Trait Alignment **a.** Scatter plots for the addiction index and module eigen-genes of both control (addiction index = 0) and addicted mice. We presented three scatter plots, where four dominant modules on each plot were selected, according to increasing, flat, decreasing correlation patterns between the module eigen-gene and the behavioral trait (addiction index). Eigengene level was calculated in profile of real samples, instead of time-series generated samples **b.** Distribution of P-value and correlation for all modules, samples, and regions. We have 51 highly correlated modules (32 positive and 19 negative correlation) by the criteria (P-value < 0.05 and | correlation coefficient value|>0.5) **c.** Trait alignment results show most of the highly correlated modules with behavioral trait belong to the NAc region.

in the NAc, which is accompanied by several gene modules in intermediate stages. These intermediate gene modules lead to gene modules that imply biological changes or shifts in addiction. Five NAc modules on day 14 including regulation of the antigen receptor-mediated signaling pathway (Module 34), were correlated with module 49. Three BLA modules on day 14 including microglial cell activation (Module 6) and response to pain (Module 9), were correlated with Module 49.

Several studies have showed the role of HSPGs in the Wnt/β-catenin/FGF signaling pathway [42], synaptic connectivity [43], and neurogenesis [44]. This time-series interplay of gene modules provides a spatiotemporal view of cocaine addiction at the transcriptome level. We applied a clustering method to the correlation matrix to elucidate other correlation patterns in this network.

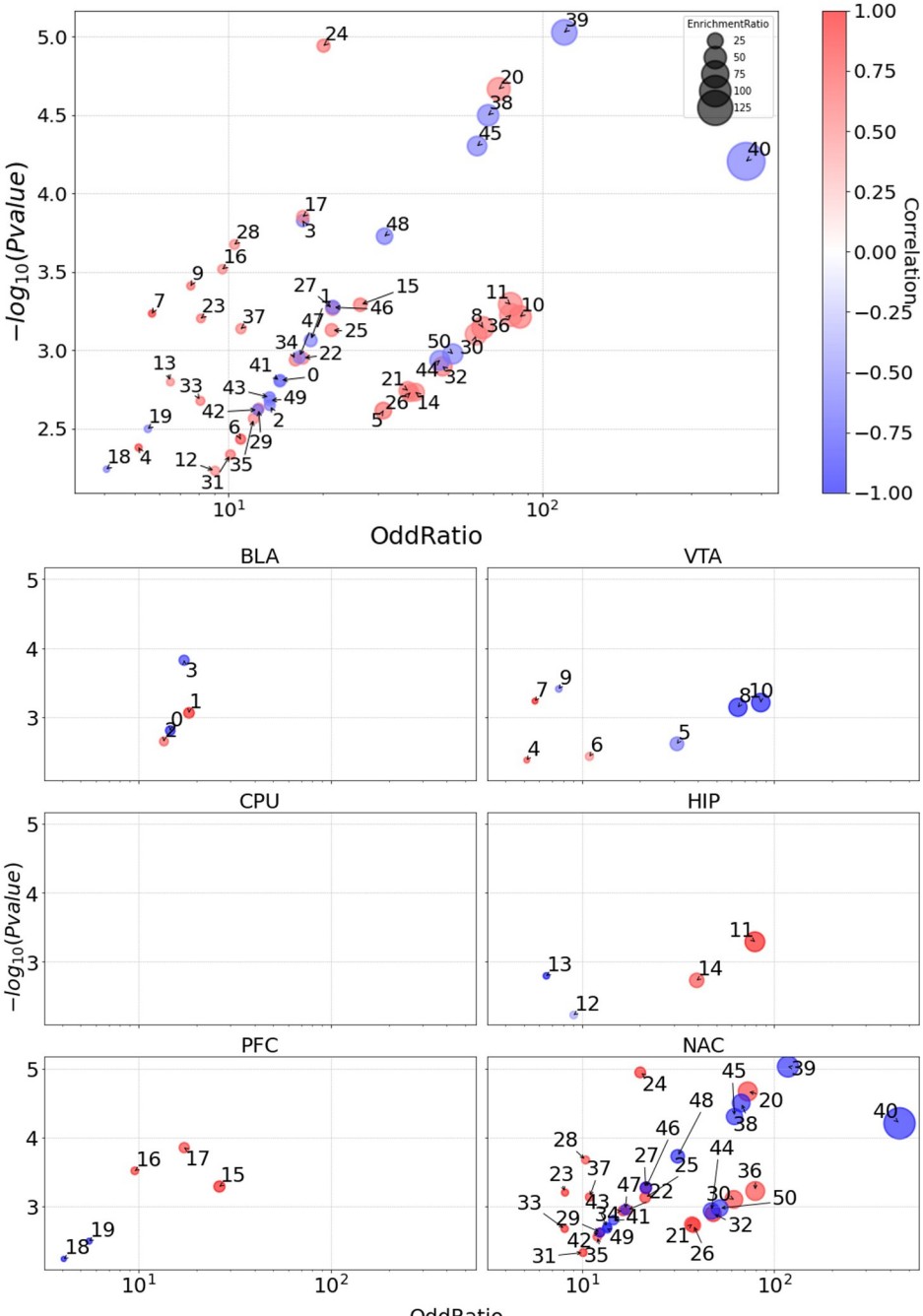

**Fig 2. GO terms of addictive behavior modules.** The primary GO terms of 51 modules (S1 Table) that highly correlated with addictive behavior were plotted with four features. P-value, Oddratio and EnrichmentRatio are evaluated from the GO enrichment analysis and correlation coefficients are calculated from the WGCNA module-trait alignment. The bottom panel figures show the modules by regions.

Among the highly addiction-behavior-correlated modules in five brain regions, excluding the CPU that does not have any highly correlated modules, a unique correlation pattern was found in the temporally aligned network of the NAc region. S7 Fig shows the quantitative attributes of the correlation patterns. First, the modules were grouped into early, mid, or late based

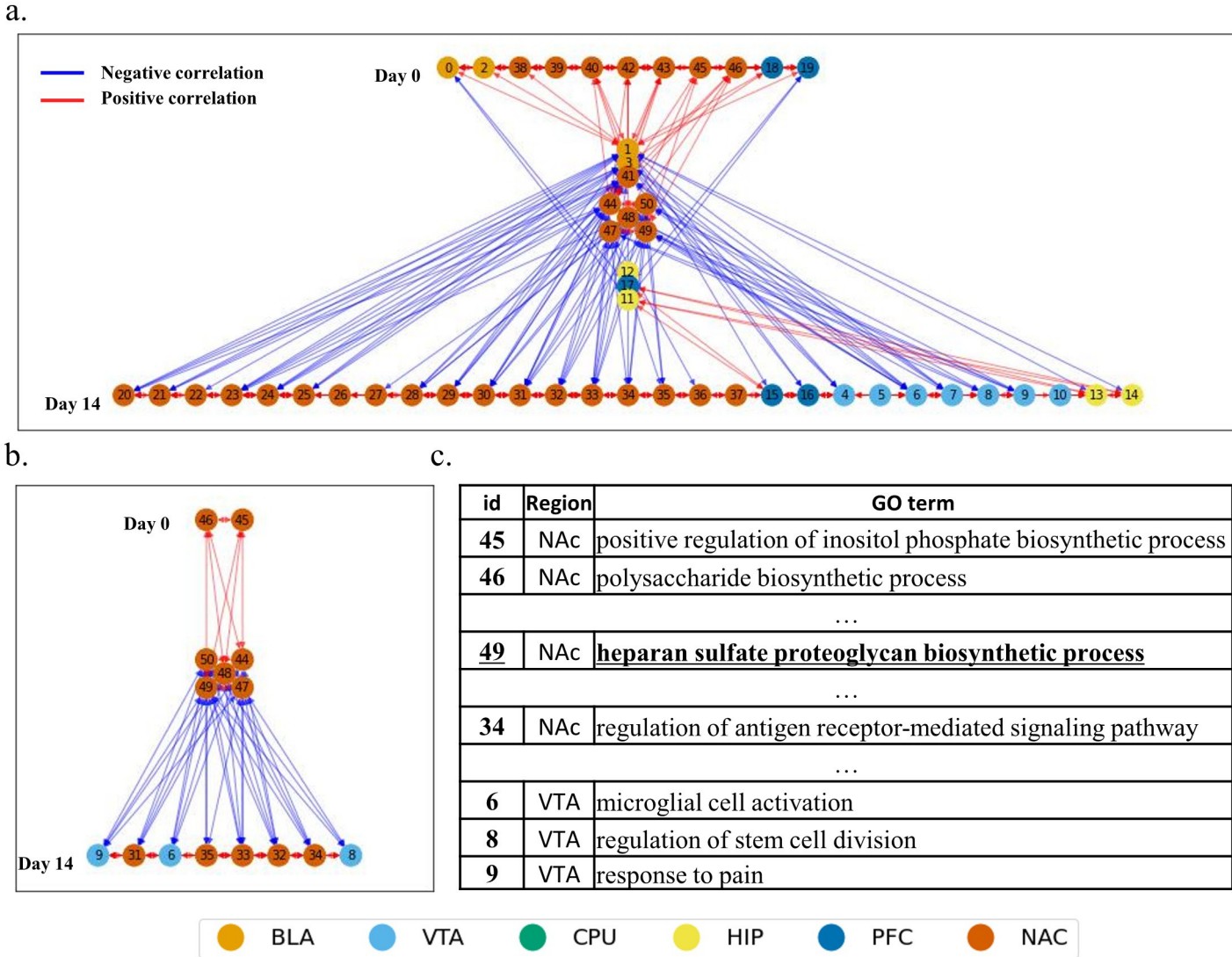

**Fig 3. Temporally Aligned Module Network a.** Each module is aligned to a specific timepoint where its module eigengene has the highest value. **b.** Visualization of regional and inner regional correlation patterns. Red lines means positive correlation and blue lines means negative correlation. **c.** GO terms of intermediate gene module which correlated with HSPG biosynthesis process.

on their aligned temporal positions. Second, the grouped modules were annotated as 1 or 2 based on their distinctive correlation patterns, which were distinguished by visual inspection and hierarchical clustering of module-to-module connections.

Mid1 and mid2 had positive correlations between early stage modules. Mid1 had a non-specific negative correlation with late-stage modules, including late1, in comparison to mid2, which had a specific negative correlation with late2. These distinctive correlations suggest the existence of different biological processes during cocaine addiction (Fig 4A and 4B). Indeed, each group in the NAc region exhibited different GO results. Late1 and late2 both had nervous system-related GO terms, but there were unique GO terms belonging to each group (Fig 4C).

The unique GO terms of the late1 group were locomotion (GO:0040011), cell motility (GO:0048870), small molecule metabolic processes (GO:0044281), and localization of cell migration (GO:0051674), which suggests vascular or glia-related biological processes (S8A and

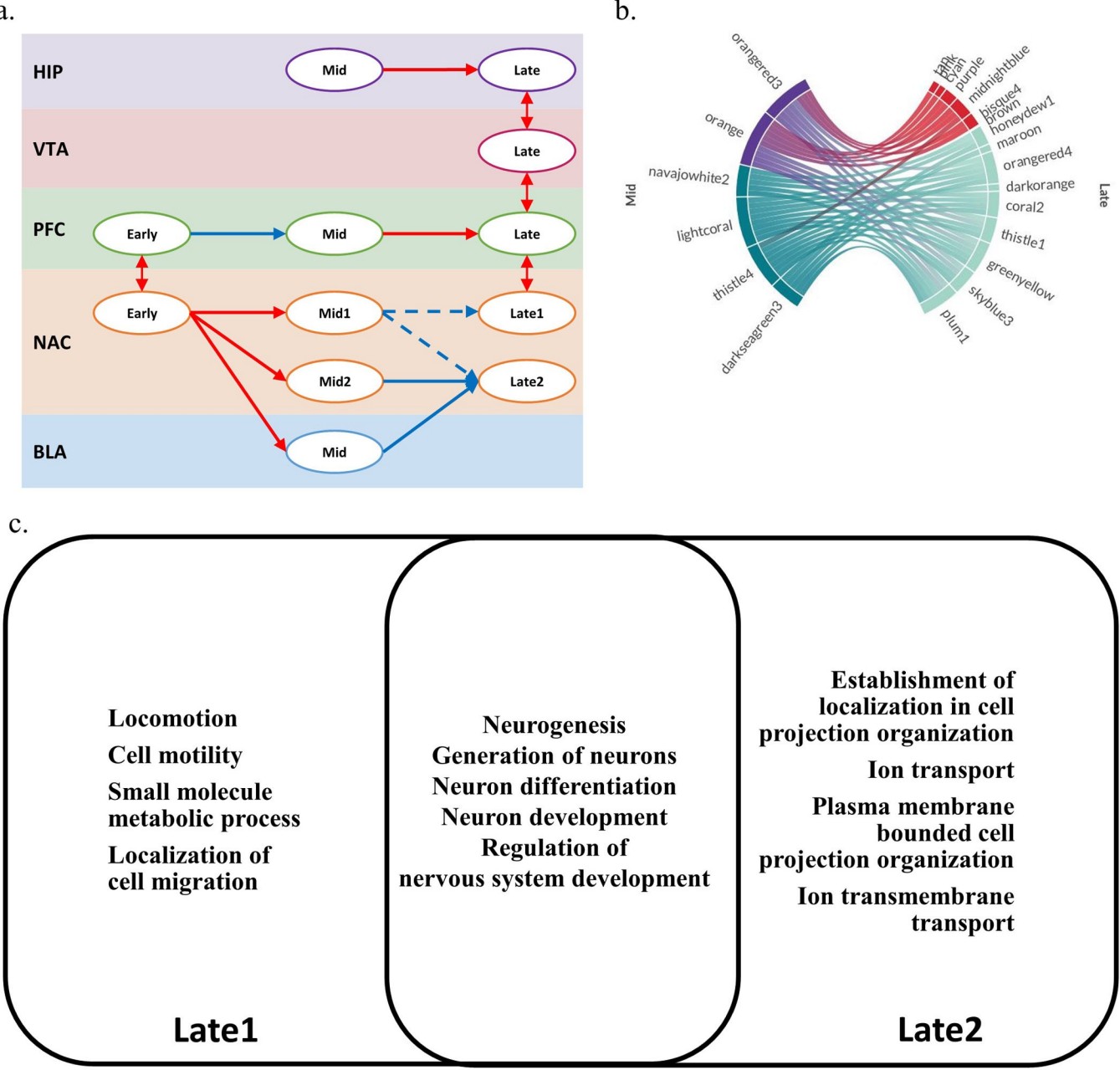

**Fig 4.** Specific Correlation Pattern Shows Distinct Differences Between Cell Migration and Membrane- Related Biological Process **a.** Diagram of correlation pattern **b.** Visualization of correlation pattern between mid and late stage modules **c.** Gene ontology analysis of separated module group shows not only common GO terms but also distinct differences, which suggest different biological processes are involved in these correlation patterns.

S8B Fig). In the late2 group, the establishment of localization in cell projection organization (GO:0051649), ion transport (GO:0006811), plasma membrane-bounded cell projection organization (GO:0120036), and ion transmembrane transport (GO:0034220) were included. These results suggest membrane- or neuron-related biological processes. Mid1 and mid2 both showed regulatory-related GO terms. (S8A and S8B Fig) Module 49 that was denoted as a HSPG related module was included in mid2 group. Regulatory GO terms in the mid2 group appear to be consistent with this result.

In summary, we enabled qualitative screening of intermediate key gene modules and genes that show correlation with addictive behavior, both cross-checked in literature and validated in a well-accepted manner using conventional WGCNA methods. This approach ensures the robustness and relevance of our findings in the context of spatiotemporal study including addiction research.

## Rescued DEG provides statistically significant gene expression analysis in intermediate modules

There are two reasons why DEG cannot be used in these findings. First, the averaged gene expression profile used in previous studies does not provide biological replicas, which leads to a lack of statistical significance calculations. Second, genes included in the intermediate modules, whose highest point of log2FC was located in the intermediate module. Therefore, they can be excluded from conventional DEG because of the lack of sufficient log2FC and high probability because conventional DEG only compares control (start point) and treatment (end point), not intermediate.

To overcome this limitation and provide statistical metrics for gene expression profile interpretation, we devised a rescued DEG (rDEG), which is an extended version of the DEG. Figs 5, S9A and S9B show the basic concept of rDEG, which is a serial DEG calculation analysis using a raw expression profiles as input instead of the averaged gene expression profile that was originally used. rDEG not only provides a statistical metric–P-value–but also "rescues" possibly missed DEGs because of going up and down gene expression patterns in the middle, as shown in Fig 5.

Indeed, rDEGs were calculated from intermediate modules that were originally missed by conventional DEGs and showed biological validity in both GO analysis and literature studies (S2 Table). By screening genes that were included in significant GO terms and filtering significant genes using rDEG criteria, we showed the important time series interplay of genes that have statistical significance. Considering their regulatory GO terms, the rDEGs in the intermediate module could be intermediate regulators that were missed in a previous study.

We searched for rDEGs based on previous studies related to addiction or synaptic plasticity (S2 Table). Modules 44, 47, 48, 49, and 50 have several rDEGs and corresponding references, which implies inter-module co-expression related to HSPG (module 49). There were also other rDEGs not included in the mid2 group but included in the mid1 group–Celf in module 41. The significance of these intermediate rDEGs can be shown by a brief introduction to previous research: Alcam for mid2 and Celf for mid1.

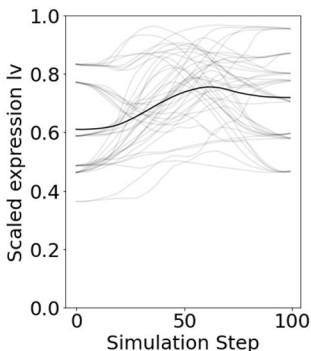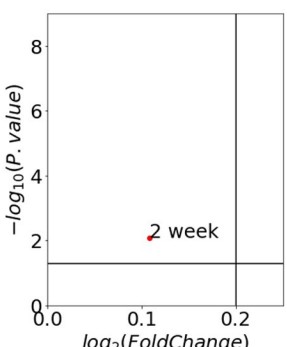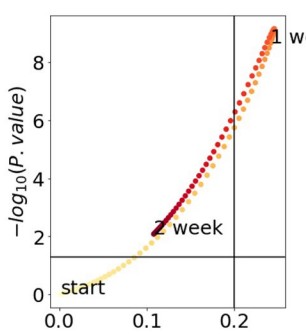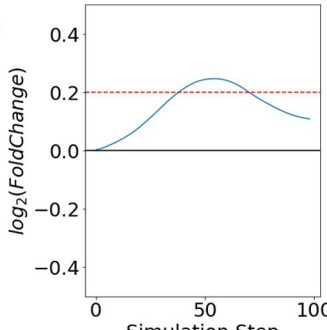

**Fig 5. Rescued DEG (rDEG) and its validation an example of rDEG (Zfp51 in NAc), which originally sorted as a non-significant gene because of lack of fold change, but was rescued in rDEG.**

An example of an important time series interplay of genes is activated leukocyte cell adhesion molecule (Alcam), which is known to modulate midbrain dopamine neurons (mDA). Dysfunction of the mDA circuitry is related to neurodegenerative and neuropsychiatric conditions, including drug addiction [45]. In the time-series view of cocaine addiction, Alcam shows its significance in intermediate stages and mediates membrane-related modification in gene module 48, which includes group mid2: neuron development, neuron differentiation, regulation of nervous system development, cell projection organization, plasma membrane-bound cell projection organization, ion transport, ion transmembrane transport, generation of neuronal neurogenesis, and establishment of localization in cell and ion transport ion transmembrane transportation (S9C Fig). Because of the detailed view of the late2 gene group, GO analysis using rDEG showed which specific genes were related to behavioral features of cocaine addiction in focal adhesion, and the collagen-activated signaling pathway. Alcam, a cytoplasmic membrane protein, provides a complete time-series view of membrane-related modifications during cocaine addiction

Another example that implies an important interplay in gene modules 41 and mid1 group is mediated by CUGBP Elav-like family member 4(Celf4), which includes RNA processing (GO:00006396) and rDEG. Celf4 is known to play a role in neuronal differentiation and excitation, corticothalamic development, synaptic transmission and function, and synaptic plasticity [46]. Celf4 regulates local translation, including a vast set of mRNAs associated with the regulation of synaptic function. Considering that the mid1 eigengene profile shows a weak negative correlation with the late1 and late2 groups of gene modules that are not only highly correlated with behavioral features but also placed in the intermediate stage of the addiction process, Celf4 might be important for the cocaine addiction process. Recent research that implies a relationship between Celf4 and amphetamine addiction [47] also supports the possible connection between Celf4 and cocaine addiction.

In addition, although there is no direct evidence for this, a prediction can be made based on indirect evidence that describes the interaction between HSPG and cadherin [48–51]–Cdh11 can be a possible therapeutic target for cocaine addiction, which is not only one of the rDEGs of the HSPG biosynthetic process module (module 49), but also a well-known transmembrane proteins that mediate cell–cell adhesion in signaling.

To assist in the interpretation of our findings, more extensive reviews on spatiotemporal aspects of cocaine addiction and implications are described in S4 File. Furthermore, an explanation focusing on differences between WGCNA and GAN-WGCNA can be found in S5 File.

In summary, we enabled quantitative measurement of intermediate key genes identified in previous analyses by applying conventional Differentially Expressed Gene (DEG) methods to the generated time-series gene expression profiles. This approach allowed us to validate the significance of these genes in a dynamic context, providing a more detailed and robust understanding of their roles in well-acceptable manner.

## Discussion

Machine learning is a promising computational method that can be used to overcome existing limitations in many domains, including biological research [52]. However, its interpretation and assessment are challenging, especially in the biological domain [53]. We suggest a pipeline that enables not only the utilization of augmented information from the GAN, but also the assessment of the statistical significance of generated gene expression data. Because our methods are an extension of conventional methods, they are easily glued to other methods by exchanging specific algorithms or adding extra analytic pipelines.

We proposed a GAN-WGCNA method using generated gene expression data, which is an unbiased systemic approach for the full utilization of high temporal resolution. Our method, which combines GAN and WGCNA, improved its analytic results and enabled not only unbiased analysis but also calculation of the correlation between temporal gene modules and addiction behavior. The key idea of combining GAN and WGCNA is to detect gene modules in the generated time-series profile of each gene and calculate their eigen gene profiles in real samples that have trait information. Indeed, these approaches showed that gene modules in the NAc are mostly involved in addiction processes, consistent with previous studies [26, 36, 54, 55]. We rescued DEGs in the intermediate stages of cocaine addiction using the rDEG method. This enables the understanding of time-series biological processes. GAN-WGCNA combined with rDEG, has the capability of unbiased and intermediate-level analysis, that allowed the rescue of at least two important genes in the intermediate stage.–Celf4 [46, 47], and Alcam [45].

This study had a few limitations. One is that the aligned time point was determined by the peak point of the expression profile. Most genes belong to the very first or last time points because their expression profiles simply decrease or increase, making it meaningless to some extent. For these genes, using differential values instead of the peak point will be more meaningful. This differential value implies the gene module or biological process that shows rapid changes during cocaine addiction. Therefore, it cannot fully guarantee its biological association with a specific time point. We expect more delicate alignment methods to enable sophisticated and intuitive interpretations. These advanced alignment methods could be based on several biological principles or mathematical calculations, such as differential or integral, which can reflect transitional or accumulated changes, respectively.

Another limitation is the gene module calculation step in WGCNA. Because of the threshold problem, only the averaged gene expression profile is currently available as an input. Other inputs, such as the unaveraged profile, did not satisfy the scale-free topological threshold, which was established empirically using experimental data. An alternative algorithm or principle is required for generated data that have a higher resolution or dimensionality compared to traditional experimental data.

Currently, biological researchers are only able to understand a fraction of data using insufficient dimensional reduction methods and face endless endeavors to reconstruct the entire map of biological systems from scattered puzzle pieces. Therefore, methods that enable efficient and significant use of machine learning technology in biological research should be devised to help researchers construct a richer biological hypothesis, which will lead to breakthroughs in understanding complex biological phenomena.

## Conclusion

We proposed generative adversarial networks and weighted gene co-expression network analysis (GAN-WGCNA) pipeline and rescued differentially expressed gene(rDEG) methods to provide an analysis for intermediate regulators. GAN can be used to generate time-series gene expression profiles without any intermediate sample collection between control and pathological state. WGCNA is an un-biased systemic level analysis method which suitable to analyze generated time-series gene expression profiles. Combination of these two methods, GAN-WGCNA indeed provides information about time-series interplay of gene modules during cocaine addiction. Moreover, rDEG rescued two intermediate regulators (Alcam and Celf4) of cocaine addiction which were missed in previous research. Considering that Alcam and Celf4 have known correlations with addiction-related behaviors. Alcam is involved in cell-cell adhesion, which is crucial for synaptic plasticity and neural connectivity, potentially influencing the neural circuits involved in addiction. Celf4 regulates RNA processing and local

translation, which are critical for synaptic function and plasticity. These findings suggest that GAN-WGCNA is capable of capturing intermediate process from bulk dataset in both inter-cellular and intracellular phenomena.

In summary, understanding intermediate regulators in spatiotemporal pathogenesis is essential to treatment and individual components of this pipeline are already well accepted methods and provide qualitative and quantitative windows to biological researchers, presented work will contribute to various future research and drug development.

## Supporting information

**S1 Fig.** GAN training details a. Table of Hyper-parameters used during GAN training b. detailed parameters for data dimension transitions; Ntrain is the number of linear augmented samples Nsim is the number of simulations which has done between CN and SN, Ngex is number of genes which is differ among brain regions c. training validation through overlapping effect in tSNE space during GAN training.
(PDF)

**S2 Fig.** Research Workflow Transitions of dataset's shape during preparing training data, model training, simulating and GAN-WGCNA a. De-scription of a used dataset (GSE110344). The tissue collection performed in 6 brain regions b. Training data preparation for GAN train-ing. Simple 10-fold linear aug-mentation method is applied. c. GAN training framework struc-ture d. Gene expression simulation through latent space interpolation using a trained generator model e. GAN-WGCNA using adjacency matrix from averaged gene profile and behavioral datasets from original samples which provide a spatiotemporal analysis of the tran-scriptome f. Rescued DEG used to show statistically significant gene expression profile.
(PDF)

**S3 Fig. Creb family genes expression profile Creb gene family expression pattern shows spatiotemporal differences which enables detailed interpretation of cocaine addiction and sophisticated module detection in WGCNA.**
(PDF)

**S4 Fig.** WGCNA a. Selection of Soft threshold. Note that couldn't meet conventional criteria in NAc region b. Module was detected after calculation of Topological overlapped Matrix and Dynamic Tree Cut method which is basically cut-off dendrogram on specific height and con-ditions.
(PDF)

**S5 Fig. Eigen-gene profiles of dominant four selected modules in order of the addition index of original samples the eigen-gene expression profile, which is calculated in gener-ated dataset, gives us an opportunity of calculating correlation to the additive behavior in temporal aspect i.e. addiction progress.**
(PDF)

**S6 Fig. Temporally aligned module network (full) network visualization to show full con-nection including day0 and day14 connection.**
(PDF)

**S7 Fig. Clustering of correlation matrix to distinguish correlation patterns in modules spe-cific correlation pattern was founded in early-mid-late stage of NAc region.**
(PDF)

**S8 Fig.** GO results a. GO result of Mid1, Mid2, Late1, Late2 groups in NAc. b. Comparison of GO DAG between Late1 and Late2.
(PDF)

**S9 Fig.** rDEG visualization of Alcam, Celf4 and Cdh11 a. Visualized examples are intermediate example that are possible DEG in intermediate stages b. rDEG-filtered GO result shows detailed term comparing to the non-filtered result c. Comparison between DEG and rDEG shows rDEG is an extended version of DEG.
(PDF)

**S10 Fig. Spatiotemporal expression patterns of behavioral related modules three visualized examples for each behavior related gene modules in different brain regions–module 48 in NAc, module 16 in PFC, and module 10 in VTA.** Six representative genes were selected for visualization.
(PDF)

**S11 Fig.** Comparison between WGCNA and GAN-WGCNA a. Differences of WGCNA and GAN-WGCNA in three aspects b. Distinguishing features of GAN-WGCNA in identifying and computing eigengenes across different samples.
(PDF)

**S1 Table. GO table of gene module modules' GO terms and correlation calculation results.** Corr and P.val calculated from WGCNA module-trait alignment and G.Pv(Pvalue) and OR (OddRatio) and ER(EnrichmentRatio) calculated from GO analysis.
(PDF)

**S2 Table. Genes in intermediate modules notable genes in intermediate modules (41, 44, 47, 48, 49, 50) and its references.**
(PDF)

**S1 File. C57BL/6J cocaine self-administration dataset.**
(PDF)

**S2 File. Voom and data preprocessing for training.**
(PDF)

**S3 File. Weighted gene co-expression network analysis.**
(PDF)

**S4 File. Spatiotemporal aspects of cocaine addiction and implications based on results.**
(PDF)

**S5 File. Summary and comparison between WGCNA and GAN-WGCNA.**
(PDF)

## Acknowledgments

The authors thank the Daegu Gyeongbuk Institute of Science & Technology supercomputing and big data center for allocating us with dedicated supercomputing time.

## Author Contributions

**Conceptualization:** Mookyung Cheon, Wookyung Yu.

**Data curation:** Taehyeong Kim, Kyoungmin Lee.

**Formal analysis:** Taehyeong Kim.

**Funding acquisition:** Mookyung Cheon, Wookyung Yu.

**Investigation:** Kyoungmin Lee.

**Methodology:** Taehyeong Kim, Kyoungmin Lee.

**Project administration:** Mookyung Cheon, Wookyung Yu.

**Software:** Taehyeong Kim.

**Supervision:** Mookyung Cheon, Wookyung Yu.

**Validation:** Taehyeong Kim.

**Visualization:** Taehyeong Kim.

**Writing – original draft:** Taehyeong Kim, Kyoungmin Lee, Wookyung Yu.

**Writing – review & editing:** Taehyeong Kim, Kyoungmin Lee, Mookyung Cheon.

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
