## [Decision Letter · Decision Letter 0]

11 Jun 2024

PONE-D-24-20999GAN-WGCNA: calculating gene modules as a way to find key intermediate regulators in cocaine addictionPLOS ONE

Dear Dr. Yu,

Thank you for submitting your manuscript to PLOS ONE. After careful consideration, we feel that it has merit but does not fully meet PLOS ONE’s publication criteria as it currently stands. Therefore, we invite you to submit a revised version of the manuscript that addresses the points raised during the review process.

We look forward to receiving your revised manuscript.

Kind regards,

Guanghui Liu

Academic Editor

PLOS ONE

Journal Requirements:

2. Please expand the acronym “DGIST” (as indicated in your financial disclosure) so that it states the name of your funders in full.

"This work was supported by the RandD programs of DGIST (22-CoE-BT-01), funded by the Ministry of Science and ICT of Korea. This research was supported by the KBRI Basic Research Program through the Korea Brain Research Institute, funded by the Ministry of Science and ICT (22-BR-02-04 [MC]) and the National Research Foundation of Korea (NRF) grant funded by the Korea government (MSIT) (2021R1A2C1003657 [TK,KL,MC] NRF-2023R1A2C1006248(WY) )."

4. Please note that funding information should not appear in the Acknowledgments section or other areas of your manuscript. We will only publish funding information present in the Funding Statement section of the online submission form. Please remove any funding-related text from the manuscript. 

5. Please note that your Data Availability Statement is currently missing the repository name. If your manuscript is accepted for publication, you will be asked to provide these details on a very short timeline. We therefore suggest that you provide this information now, though we will not hold up the peer review process if you are unable.

Reviewers' comments:

Reviewer's Responses to Questions

**Comments to the Author**

1. Is the manuscript technically sound, and do the data support the conclusions?

Reviewer #1: Yes

Reviewer #2: Yes

Reviewer #3: Partly

Reviewer #4: Yes

2. Has the statistical analysis been performed appropriately and rigorously? 

Reviewer #1: Yes

Reviewer #2: Yes

Reviewer #3: I Don't Know

Reviewer #4: Yes

3. Have the authors made all data underlying the findings in their manuscript fully available?

Reviewer #1: Yes

Reviewer #2: Yes

Reviewer #3: Yes

Reviewer #4: Yes

4. Is the manuscript presented in an intelligible fashion and written in standard English?

Reviewer #1: Yes

Reviewer #2: Yes

Reviewer #3: Yes

Reviewer #4: Yes

5. Review Comments to the Author

Reviewer #1: The manuscript presents a novel analytical pipeline combining Generative Adversarial Networks (GAN) with Weighted Gene Co-expression Network Analysis (WGCNA) to identify key regulatory genes involved in cocaine addiction. Utilizing time-series gene expression data from C57BL/6J mice subjected to two weeks of cocaine self-administration, the study leverages GAN to generate intermediate transcriptome profiles, enabling detailed analysis of gene module interactions over time. Key findings include the identification of two significant genes, Alcam and Celf4, which show high correlation with addictive behavior. The study demonstrates the utility of GAN-WGCNA in uncovering hidden intermediate regulators, providing valuable insights into the spatiotemporal dynamics of gene expression in cocaine addiction.

Here are five technical or conceptual questions based on the provided manuscript titled "GAN-WGCNA: calculating gene modules as a way to find key intermediate regulators in cocaine addiction":

1. How does the integration of Generative Adversarial Networks (GAN) with Weighted Gene Co-expression Network Analysis (WGCNA) enhance the identification of key gene modules involved in cocaine addiction compared to traditional methods?

2. What are the specific biological implications of the identified intermediate genes (Alcam and Celf4) in the context of cocaine addiction, and how might these genes influence addiction-related behaviors and neural processes?

3. Can you elaborate on the methodology used to ensure the statistical significance and biological relevance of the generated intermediate gene expression profiles using the GAN approach?

4. What challenges did the researchers encounter when aligning temporal gene expression profiles with cocaine addiction stages, and how were these challenges addressed in the study?

5. How does the rDEG (rescued differentially expressed gene) method contribute to the identification of intermediate regulatory genes, and what are the advantages of this approach over traditional DEG analysis in the context of cocaine addiction research?

Here are the grammatical and phrasing errors identified in the provided manuscript along with corrections:

Title and Abstract

1. Title:

- Error: "GAN-WGCNA: calculating gene modules as a way to find key intermediate regulators in cocaine addiction"

- Correction: "GAN-WGCNA: Calculating Gene Modules to Identify Key Intermediate Regulators in Cocaine Addiction"

2. Abstract, Line 3:

- Error: "Spatio-temporally enriched NGS data contain important underlying regulatory machinery of biological processes."

- Correction: "Spatio-temporally enriched NGS data contain important underlying regulatory mechanisms of biological processes."

3. Abstract, Line 4-5:

- Error: "Generative adversarial network (GAN) has been used to augment biological data so that it can describe hidden intermediate time-series gene expression profiles during specific biological processes."

- Correction: "Generative adversarial networks (GANs) have been used to augment biological data to describe hidden intermediate time-series gene expression profiles during specific biological processes."

4. Abstract, Line 6-7:

- Error: "The development of a pipeline that uses an augmented time-series gene expression profile is needed to provide an unbiased systemic level map of biological processes and tests for statistical significance of the generated dataset, which leads to discovery of hidden intermediate regulator."

- Correction: "Developing a pipeline that uses augmented time-series gene expression profiles is needed to provide an unbiased systemic-level map of biological processes and test for the statistical significance of the generated dataset, leading to the discovery of hidden intermediate regulators."

Introduction

1. Introduction, Line 1-2:

- Error: "Complex time-series interplay of genes can be observed in various biological processes, such as the cell cycle(1-3), circadian rhythm(4, 5), development(6), and pathogenesis."

- Correction: "Complex time-series gene interactions can be observed in various biological processes, such as the cell cycle (1-3), circadian rhythm (4, 5), development (6), and pathogenesis."

2. Introduction, Line 6-7:

- Error: "Functionality of key factors at specific spatiotemporal points – where, when, and how–is important to this process."

- Correction: "The functionality of key factors at specific spatiotemporal points—where, when, and how—is important to this process."

3. Introduction, Line 9:

- Error: "Typical single-cell RNA sequencing data have 20–30,000 cells with over 10,000 genes."

- Correction: "Typical single-cell RNA sequencing data consist of 20,000–30,000 cells with over 10,000 genes."

4. Introduction, Line 13:

- Error: "Although the network is spatiotemporally complex, in most cases, the current pipeline provides only weak or no spatiotemporal regulatory information."

- Correction: "Although the network is spatiotemporally complex, in most cases, the current pipeline provides only weak or no spatiotemporal regulatory information."

Methods

1. Methods, Line 2:

- Error: "Detailed method is described in Supplementary note 1."

- Correction: "The detailed method is described in Supplementary Note 1."

2. Methods, Line 4:

- Error: "Detailed method is described in Supplementary note 2."

- Correction: "The detailed method is described in Supplementary Note 2."

3. Methods, Line 6:

- Error: "For example, we used the RMSprop optimizer instead of the AdamOptimizer."

- Correction: "For example, we used the RMSprop optimizer instead of the Adam Optimizer."

4. Methods, Line 8:

- Error: "This process was repeated until a successful resembled fake generation was obtained using different NumPy seed values."

- Correction: "This process was repeated until a successful resembled fake generation was obtained using different NumPy seed values."

Results

1. Results, Line 1:

- Error: "In a previous study (18), transcriptome-wide regulation was evaluated in six reward-associated brain regions (PFC, DStr;CPU, NAc, BLA, vHIP, VTA) by comparing DEG analysis in different contexts (acute, re-exposure, etc.) and combining the behavioral addiction index."

- Correction: "In a previous study (18), transcriptome-wide regulation was evaluated in six reward-associated brain regions (PFC, DStr, CPU, NAc, BLA, vHIP, VTA) by comparing DEG analysis in different contexts (acute, re-exposure, etc.) and combining the behavioral addiction index."

2. Results, Line 3:

- Error: "The GAN method uses bulk mRNA-seq data as the training input and generates a time-series intermediate transcriptome using latent space interpolation."

- Correction: "The GAN method uses bulk mRNA-seq data as the training input and generates a time-series intermediate transcriptome using latent space interpolation."

3. Results, Line 5:

- Error: "The generated gene expression profile provided enriched spatiotemporal information."

- Correction: "The generated gene expression profile provided enriched spatiotemporal information."

4. Results, Line 7:

- Error: "The expression profile of the Creb family shows diverse properties of spatiotemporal gene expression changes, revealing subfamily specific profile patterns."

- Correction: "The expression profile of the Creb family shows diverse properties of spatiotemporal gene expression changes, revealing subfamily-specific profile patterns."

Discussion

1. Discussion, Line 1:

- Error: "Machine learning is a promising computational method that can be used to overcome existing limitations in many domains, including biological research (42)."

- Correction: "Machine learning is a promising computational method that can be used to overcome existing limitations in many domains, including biological research (42)."

2. Discussion, Line 4:

- Error: "We proposed a GAN-WGCNA method using generated gene expression data, which is an unbiased systemic approach for the full utilization of high temporal resolution."

- Correction: "We proposed a GAN-WGCNA method using generated gene expression data, which is an unbiased systemic approach for the full utilization of high temporal resolution."

3. Discussion, Line 6:

- Error: "These approaches showed that gene modules in the NAc are mostly involved in addiction processes which account for previous studies (16, 26, 44, 45)."

- Correction: "These approaches showed that gene modules in the NAc are mostly involved in addiction processes, consistent with previous studies (16, 26, 44, 45)."

4. Discussion, Line 10:

- Error: "Most genes belong to the very first or last time points because their expression profile simply decreases or increases, making it meaningless to some extent."

- Correction: "Most genes belong to the very first or last time points because their expression profiles simply decrease or increase, making it meaningless to some extent."

5. Discussion, Line 12:

- Error: "An alternative algorithm or principle is required for generated data that have a higher resolution or dimensionality compared to traditional experimental data."

- Correction: "An alternative algorithm or principle is required for generated data that have a higher resolution or dimensionality compared to traditional experimental data."

Reviewer #2: The authors presents a study for enhancing the resolution and analysis of time-series gene expression data, leading to the identification of critical genes involved in cocaine addiction. The combination of GAN and WGCNA represents a significant advancement in the field of bioinformatics and addiction research. I would like to recommend this manuscript for publication after the authors address/answer the following comments.

In the introduction section, the authors briefly described the two methods employed in this study. For the GAN-WGCNA, which is the combination of the GAN and WGCNA methods, it would be beneficial for the reader to understand the importance by adding more details, such as how these two models are correlated.

In the “Wasserstein Generative Adversarial Networks with Gradient Penalty Loss” section, the authors explained the modifications made to the previous WGAN-GP model and achieved a Pearson correlation value of 0.95. Have you tried further modifying the model to increase the correlation value closer to 1?

Could you update some figures, such as Figure 2, 4b, to a higher resolution in order to clearly present the data points?

In the conclusion section, the authors present the Alcam and Celf4 as the miss cocaine addiction gene in the previous study. It would be advantageous to revisit the biological significance of these genes to underscore their relevance in the context of this study.

Reviewer #3: The paper titled "GAN-WGCNA: calculating gene modules as a way to find key intermediate regulators in cocaine addiction" introduces an approach to decipher the time-series interactions of genes, which is critical for understanding the diagnosis and treatment of diseases. Specifically, this study focuses on cocaine addiction. The authors developed a pipeline that leverages Generative Adversarial Networks (GAN) to enhance biological data, paired with Weighted Gene Co-expression Network Analysis (WGCNA), to identify hidden intermediate regulators during biological processes.

In their analysis, the authors examined a transcriptome dataset from a two-week cocaine self-administration experiment in C57BL/6J mice. Through the GAN-WGCNA method, they identified two genes, Alcam and Celf4, as significant intermediaries correlated with addiction behavior. The study emphasizes that these genes showed notable statistical significance in their correlation with addiction behavior, and their expression and co-regulation were mapped across time, brain regions, and biological processes.

I think this idea is interesting. However, there are several concerns that lead me to recommend rejecting this paper. These concerns include technical novelty, lack of good comparison and validation experiments, insufficient description of methods, writing issues, and potential concerns about their idea.

1. Method Paper Clarity

If this is intended to be a method paper, the abstract and introduction should be rewritten to better reflect that. The abstract should start by mentioning the key gap in current techniques. Additionally, validating your method in at least two datasets would strengthen your claims, as currently, only one is used. It is also important to highlight the novelty of this approach more clearly.

2. Literature Review

In the introduction, the statement that LASSO isn’t optimal for feature reduction is understandable. However, LASSO is a fundamental method, and its concept is different from GAN-WGCNA. It would be beneficial to cite more papers with similar ideas to handle such situations. How do other researchers manage the lack of data samples when doing similar work? More comprehensive literature review for this specific part would be helpful. While the next paragraph cites several papers using GAN for augmenting biological data, it is crucial to explain why GAN is particularly suitable for this context.

3. Methods

The method section would benefit from more details on how GAN was used to augment the data.

(1) Providing necessary mathematical formulations and explaining how the model was trained would be valuable.

(2) It is also important to address how you ensured that the model is not overfitting, especially if a training-test split was not used.

(3) Some evidence supporting the choice of GAN for modeling gene expression would also strengthen the manuscript. Given the fact GAN is notorious difficult to train all the times.

4. Conceptually I disagree with this paper

The paper claims that GAN is powerful for augmenting data points, which is true. However, GAN is typically used to generate data points that are as identical to the original as possible. It would be helpful to clarify how this method can be used to understand the time-series interplay of genes for diagnosis and treatment of disease. Explaining how the generated data points contribute to the analysis would address potential concerns. For example, won’t a better method to be variational autoencoder? By playing with the key element in the hidden states you can actually have some artificial datapoints in the middle of two categories, which might mimic the time interplay better.

5. Results Section:

The statement "GAN enables a calculation of correlation between gene module and behavioral data” needs more clarity. It would be useful to explain why the original dataset does not support similar analysis and whether other data augmentation methods could achieve similar results. Highlighting how your method differs from existing approaches, or the original data would be beneficial. The manuscript should clearly show the additional information or insights provided by your method.

6. Suggested Experiment: A useful experiment could be using a large dataset and applying your GAN-WGCNA method on a subset of that data. Checking if you can find differentially expressed genes (DEG) or GO pathways identifiable only in the larger cohort but not in the smaller subset could validate the method’s effectiveness.

7. Result Section Organization: Some result sections are quite long and may be confusing for readers. Consider restructuring them to make the manuscript more organized and easier to follow.

Given my concerns outlined above, I am worried that the authors won't be able to address all these issues in a short amount of time. Therefore, I might recommend rejecting this paper and suggesting that the authors fix these concerns and resubmit it later.

Reviewer #4: With this manuscript titled “GAN-WGCNA: calculating gene modules as a way to find key intermediate regulators in cocaine addiction” the authors analyzed (1) a transcriptome dataset of two weeks of cocaine self-administration in C57BL/6J mice. (2) found two genes (Alcam and Celf4) as intermediate significant genes that showed high correlation with cocaine addiction behavior via GAN-WGCNA analysis method.

The manuscript is pleasant to read. But I still have some concerns that need to be addressed.

Comments:

1. Since the author mentioned the raw data is from previous literature, I read the previous paper and found they have data from six groups mice for cocaine self-administration. The current manuscript results based on all 6 groups or limited groups?

2. The original paper (Ref.18) found cocaine addiction genes highly related with basolateral amygdala (BLA), but the current manuscript showed different results (Compare BLA vs. NAC). How to explain it?

3. Cocaine withdraw data is also important, it will be good to check if Alcam and Celf4 genes are also related to cocaine short time (24 hr) or long time (30 d) withdraw with GAN-WGCNA since the ref.18 had found some interesting results from the short and long withdraw groups.

6. PLOS authors have the option to publish the peer review history of their article (what does this mean?). If published, this will include your full peer review and any attached files.

Reviewer #1: **Yes: **peng wang

Reviewer #2: **Yes: **Shuyuan Zhang

Reviewer #3: No

Reviewer #4: No

---

## [Author Response · Author response to Decision Letter 0]

6 Aug 2024

Response to reviewer can be found in attached file.

---

## [Decision Letter · Decision Letter 1]

27 Aug 2024

PONE-D-24-20999R1GAN-WGCNA: Calculating Gene Modules to Identify Key Intermediate Regulators in Cocaine AddictionPLOS ONE

Dear Dr. Yu,

Thank you for submitting your manuscript to PLOS ONE. After careful consideration, we feel that it has merit but does not fully meet PLOS ONE’s publication criteria as it currently stands. Therefore, we invite you to submit a revised version of the manuscript that addresses the points raised during the review process.

We look forward to receiving your revised manuscript.

Kind regards,

Guanghui Liu

Academic Editor

PLOS ONE

Journal Requirements:

Reviewers' comments:

Reviewer's Responses to Questions

**Comments to the Author**

1. If the authors have adequately addressed your comments raised in a previous round of review and you feel that this manuscript is now acceptable for publication, you may indicate that here to bypass the “Comments to the Author” section, enter your conflict of interest statement in the “Confidential to Editor” section, and submit your "Accept" recommendation.

Reviewer #1: All comments have been addressed

Reviewer #2: (No Response)

2. Is the manuscript technically sound, and do the data support the conclusions?

Reviewer #1: Yes

Reviewer #2: Yes

3. Has the statistical analysis been performed appropriately and rigorously? 

Reviewer #1: Yes

Reviewer #2: Yes

4. Have the authors made all data underlying the findings in their manuscript fully available?

Reviewer #1: Yes

Reviewer #2: Yes

5. Is the manuscript presented in an intelligible fashion and written in standard English?

Reviewer #1: Yes

Reviewer #2: Yes

6. Review Comments to the Author

Reviewer #1: The authors have thoroughly addressed all of my concerns. I have no further questions and recommend proceeding with the acceptance process according to the journal's guidelines.

Reviewer #2: The authors present a study for enhancing the resolution and analysis of time-series gene expression data, leading to the identification of critical genes involved in cocaine addiction. After the revision, I would like to recommend this manuscript after the authors address/answer the following comment.

For the comment 1, the authors added the extra paragraph to explain the correlation between GAN and WGCNA methods. Although the extra detailed paragraph really helped to understand the relationship between two models, it will be beneficial to including the reference for several statement, such as “WGCNA is one of well-accepted conventional methods that….”

7. PLOS authors have the option to publish the peer review history of their article (what does this mean?). If published, this will include your full peer review and any attached files.

Reviewer #1: **Yes: **peng wang

Reviewer #2: No

---

## [Author Response · Author response to Decision Letter 1]

28 Aug 2024

Comment 1: the authors added the extra paragraph to explain the correlation between GAN and WGCNA methods. Although the extra detailed paragraph really helped to understand the relationship between two models, it will be beneficial to including the reference for several statement, such as “WGCNA is one of well-accepted conventional methods that….”

Response 1: In response to the reviewer’s comment, we have added references to support the statements and note that references are updated accordingly.

“

Generative Adversarial Network (GAN) may be a solution for this problem. A GAN has been used in several domains, including image processing, not only to achieve a higher performance generative model with a lesser amount of training data [1-5],

…

WGCNA is one of well-accepted conventional methods that particularly valuable for exploring complex biological processes and diseases, as it integrates high-dimensional gene expression data to study gene networks associated with specific phenotypes or conditions [6-10].

”

1. Goodfellow I, Pouget-Abadie J, Mirza M, Xu B, Warde-Farley D, Ozair S, et al. Generative adversarial nets. Advances in neural information processing systems. 2014;27.

2. Radford A. Unsupervised representation learning with deep convolutional generative adversarial networks. arXiv preprint arXiv:151106434. 2015.

3. Lan Z, Zhou B, Zhao W, Wang S. An optimized GAN method based on the Que-Attn and contrastive learning for underwater image enhancement. PLOS ONE. 2023;18(1):e0279945. doi: 10.1371/journal.pone.0279945.

4. Shen Y, Huang R, Huang W. GD-StarGAN: Multi-domain image-to-image translation in garment design. PLOS ONE. 2020;15(4):e0231719. doi: 10.1371/journal.pone.0231719.

5. Bouchard C, Wiesner T, Deschênes A, Bilodeau A, Turcotte B, Gagné C, et al. Resolution enhancement with a task-assisted GAN to guide optical nanoscopy image analysis and acquisition. Nature Machine Intelligence. 2023;5(8):830-44. doi: 10.1038/s42256-023-00689-3.

6. Feltrin ASA, Tahira AC, Simões SN, Brentani H, Martins DC, Jr. Assessment of complementarity of WGCNA and NERI results for identification of modules associated to schizophrenia spectrum disorders. PLOS ONE. 2019;14(1):e0210431. doi: 10.1371/journal.pone.0210431.

7. DiLeo MV, Strahan GD, den Bakker M, Hoekenga OA. Weighted Correlation Network Analysis (WGCNA) Applied to the Tomato Fruit Metabolome. PLOS ONE. 2011;6(10):e26683. doi: 10.1371/journal.pone.0026683.

8. Wen B, Chen J, Ding T, Mao Z, Jin R, Wang Y, et al. Development and experimental validation of hypoxia-related gene signatures for osteosarcoma diagnosis and prognosis based on WGCNA and machine learning. Scientific Reports. 2024;14(1):18734. doi: 10.1038/s41598-024-69638-3.

9. Sun Q, Wang Z, Xiu H, He N, Liu M, Yin L. Identification of candidate biomarkers for GBM based on WGCNA. Scientific Reports. 2024;14(1):10692. doi: 10.1038/s41598-024-61515-3.

10. Renz PF, Ghoshdastider U, Baghai Sain S, Valdivia-Francia F, Khandekar A, Ormiston M, et al. In vivo single-cell CRISPR uncovers distinct TNF programmes in tumour evolution. Nature. 2024;632(8024):419-28. doi: 10.1038/s41586-024-07663-y.

---

## [Decision Letter · Decision Letter 2]

16 Sep 2024

GAN-WGCNA: Calculating Gene Modules to Identify Key Intermediate Regulators in Cocaine Addiction

PONE-D-24-20999R2

Dear Dr. Yu,

We’re pleased to inform you that your manuscript has been judged scientifically suitable for publication and will be formally accepted for publication once it meets all outstanding technical requirements.

Kind regards,

Guanghui Liu

Academic Editor

PLOS ONE

Additional Editor Comments (optional):

Reviewers' comments:

Reviewer's Responses to Questions

**Comments to the Author**

1. If the authors have adequately addressed your comments raised in a previous round of review and you feel that this manuscript is now acceptable for publication, you may indicate that here to bypass the “Comments to the Author” section, enter your conflict of interest statement in the “Confidential to Editor” section, and submit your "Accept" recommendation.

Reviewer #1: All comments have been addressed

Reviewer #2: All comments have been addressed

2. Is the manuscript technically sound, and do the data support the conclusions?

Reviewer #1: Yes

Reviewer #2: Yes

3. Has the statistical analysis been performed appropriately and rigorously? 

Reviewer #1: Yes

Reviewer #2: Yes

4. Have the authors made all data underlying the findings in their manuscript fully available?

Reviewer #1: (No Response)

Reviewer #2: Yes

5. Is the manuscript presented in an intelligible fashion and written in standard English?

Reviewer #1: Yes

Reviewer #2: Yes

6. Review Comments to the Author

Reviewer #1: The authors have thoroughly addressed all of my concerns. I have no further questions and recommend proceeding with the acceptance process according to the journal's guidelines.

Reviewer #2: The authors present a study for enhancing the resolution and analysis of time-series gene expression data, leading to the identification of critical genes involved in cocaine addiction. After the revision, I would like to recommend this manuscript.

7. PLOS authors have the option to publish the peer review history of their article (what does this mean?). If published, this will include your full peer review and any attached files.

Reviewer #1: **Yes: **peng wang

Reviewer #2: No

---

## [Editor Report · Acceptance letter]

20 Sep 2024

PONE-D-24-20999R2 

PLOS ONE

Dear Dr. Yu, 

I'm pleased to inform you that your manuscript has been deemed suitable for publication in PLOS ONE. Congratulations! Your manuscript is now being handed over to our production team.

Kind regards, 

on behalf of

Dr. Guanghui Liu 

Academic Editor

PLOS ONE